Metoprolol rescues endothelial progenitor cell dysfunction in diabetes

Yan Lang 1
Dong Yi-fan 1
Qing Tao-lin 1
Deng Ya-ping 2
Han Xue 3
Shi Wen-jing 1
Li Jin-feng 1
Gao Fang-yuan 1
Zhang Xiao-fang 1
Tian Yi-jun 1
Dai Xiao-yu 1
Zhu Jiang-bo jiangbozhu1@163.com 1
Chen Ji-kuai cjk.smmu@hotmail.com 1
1 Department of Health Toxicology, Faculty of Naval Medicine, Second Military Medical University , Shanghai , China
2 Department of Pharmacy, Zhejiang Xiaoshan Hospital , Hangzhou , China
3 Laboratory Animal Center, Hangzhou Medical College , Hangzhou , China
Huang Zunnan
Electronic publication date: 2020 Jul 7
Publication date: 2020
Volume: 8
Electronic Location ID: e9306
Received 2020 Mar 26; Accepted 2020 May 16
Copyright: ©2020 Yan et al.
Copyright year: 2020
Copyright holder: Yan et al.
License: This is an open access article distributed under the terms of the Creative Commons Attribution License, which permits unrestricted use, distribution, reproduction and adaptation in any medium and for any purpose provided that it is properly attributed. For attribution, the original author(s), title, publication source (PeerJ) and either DOI or URL of the article must be cited.
License URL: https://creativecommons.org/licenses/by/4.0/

Keywords: Endothelial function, Beta-blockers, Endothelial progenitor cells, Angiogenesis, Diabetes

Funding: National Natural Science Foundation of China 81703627 81872660 81803643 The key project of food safety of the Science and Technology Ministry of China 2017YFC1600204 This work was supported by the grant from National Natural Science Foundation of China (81703627, 81872660, and 81803643) and the key project of food safety of the Science and Technology Ministry of China (2017YFC1600204). The funders had no role in study design, data collection and analysis, decision to publish, or preparation of the manuscript.

==============================
Added risk portended by diabetes in addition to hypertension has been related to an amplification of endothelial dysfunction. β-blockers are widely used for cardiovascular diseases and improve the endothelial function compared with a placebo. However, the effect of β-blockers on the endothelial progenitor cells (EPCs) function in diabetes is still unknown. Five β-blockers (metoprolol, atenolol, propranolol, bisoprolol, and nebivolol) were tested in EPC functional screening. Metoprolol improved EPC function significantly among the five β-blockers and was chosen for the in vivo tests in STZ induced diabetic mice. Reactive hyperemia peripheral arterial tonometry (RH-PAT) measurements were performed using the Endo-PAT2000 device in diabetic patients. Metoprolol, but not other β-blockers, improved EPC function in both tube formation and migration assay. EPC function was significantly decreased in diabetic mice, and metoprolol treatment restored damaged EPC migration capabilities and circulation EPC number. Metoprolol treatment promoted wound healing and stimulated angiogenesis in diabetic mice. Furthermore, metoprolol significantly enhanced eNOS phosphorylation and decreased O2− levels in EPCs of diabetic mice. In clinical trials, the RH-PAT index was significantly higher in metoprolol-treated versus bisoprolol-treated diabetics. Metoprolol could accelerate wound healing in diabetic mice and improve endothelial function in diabetic subjects, which may be mediated in part by improving impaired EPC function.

Introduction

Hypertension and diabetes mellitus are both well-known cardiovascular risk factors that are often comorbid. From 2011–2016, 77.1% of U.S. adults with diagnosed diabetes had high blood pressure (Muntner et al., 2018). Hypertensive patients with diabetes mellitus were also classified as having very high cardiovascular risk by the European Society of Hypertension/European Society of Cardiology Guidelines (Williams et al., 2018). Increasing evidence has demonstrated that the added risk portended by diabetes in addition to hypertension might be related to the amplification of endothelial dysfunction (Cleland et al., 2000). Furthermore, hypertension and diabetes mellitus have been identified as major independent predictors for impaired function of endothelial progenitor cells (EPCs) (Vasa et al., 2001). Thus, the possible impact on endothelial function should be taken into consideration when choosing an antihypertensive agent for diabetic patients with hypertension.

β-blockers are widely used for cardiovascular diseases, such as cardiac arrhythmias, arterial hypertension, and angina pectoris (Cruickshank, 2010; Frishman, 2003). The primary mechanism of β-blockers is in their capacity to block the β-adrenoceptors. However, several earlier observations indicated that part of the therapeutic effects shown by propranolol and metoprolol was associated with the antioxidant properties (Gomes et al., 2006; Vanhoutte & Gao, 2013). Further studies indicated that nebivolol treatment preserved endothelial-dependent vasodilatation and EPC mobilization, which were explained by the inhibition of NADPH oxidase activity and superoxide production in endothelial cells (Mason et al., 2009; Peller et al., 2015; Sorrentino et al., 2011). However, the different role of β-blockers on EPC function in diabetes with hypertension has yet to be determined.

We hypothesize that, under a diabetic state, the use of a β-blocker to reduce superoxide production maintains better EPC function than a β-blocker without that effect. To test this hypothesis, we compared the effects of five β-blockers (propranolol, metoprolol, atenolol, bisoprolol, and nebivolol) on endothelial cell function and EPC function. Moreover, we examined the effect of metoprolol on high glucose-induced EPC dysfunction, wound healing in mice, and endothelial function in diabetic patients.

Material and Methods

Reagents, cell culture, and treatment

Metoprolol, atenolol, propranolol, bisoprolol, nebivolol and D-glucose were purchased from Sigma-Aldrich (St. Louis, MO). Metoprolol (0.3 µM), atenolol (3 µM), propranolol (0.3 µM), bisoprolol (0.3 µM), nebivolol (3 nM) were dissolved in DMSO for cell experiments (chose the highest blood drug concentrations based on their pharmacokinetic parameters) (Eddington et al., 2000; Kamali et al., 1997; Le Coz et al., 1991; Spahn et al., 1984). Metoprolol was dissolved in carboxyl methyl cellulose (CMC)-Na (0.5%) for animal experiments. HUVECs were obtained from Fuheng Bio (FH1122, Fuheng Cell Center, Shanghai, China) and cultured in DMEM medium (HyClone, Logan, Utah) containing 4.5 mM D-glucose supplemented with 10% FBS, 100 U/ml penicillin, and 100 mg/ml of streptomycin. Cells in the high glucose group were incubated with a 33 mM D-glucose medium.

Determination of HUVEC and EPC function

Circulating EPCs were characterized using flow cytometry as cells that co-expressed Flk-1 and Sca-1 (BD Pharmingen, San Diego, CA). Mouse bone marrow (BM) EPCs were isolated, cultured and identified according to our previously established method (Chen et al., 2013). Migration assay and tube formation assay were used to evaluate HUVEC and BM-EPC functions. Migration was assayed by a filter membrane technique. Briefly, 5 × 104 cells were placed in the upper chamber of a 24-well Transwell plate (Corning Transwell, Lowell, MA) with an 8-µm polycarbonate membrane. VEGF (50 ng/ml, Sigma-Aldrich) was added to the culture medium placed in the lower chamber. After incubating at 37 °C for 24 h, the upper side of the filter was gently scraped with a cotton swab to remove non-migrating cells. After being stained with Hoechst 33258 (5 µM, Molecular Probes), cells that migrated into the lower chamber were determined by counting the stained nuclei using an Olympus IX71 fluorescence microscope (Olympus, Japan).

The angiogenic capacity was determined by the Matrigel tube formation assay (Li et al., 2016). Briefly, 2 × 104 HUVECs or 4 × 104 BM-EPCs were added into each well of a 96-well Matrigel (BD Biosciences, Bedford, MA) pre-coated (50 µl/well) plate and incubated for 3 h. The number of tubes was examined using an Olympus IX71 fluorescence microscope (Olympus, Japan) and ImageJ software (1.48v, NIH, USA).

Determination of ROS generation by flow cytometry and fluorescent microscopy

Dihydroethidium (DHE) (Invitrogen, Carlsbad, CA) assay is used to determine intracellular O2− levels. BM-EPCs were incubated with DHE (10−6 mol/L) for 30 min in a cell incubator. The fluorescence intensity for 10,000 events was measured using FACS (fluorescence activated cell sorting, BD). Culture plates were read at 518/605 nm in a SpectraMax M2e microplate reader (Molecular Devices, Sunnyvale, CA) and images were captured under the Olympus IX71 fluorescence microscope (Olympus, Japan).

Animal studies

Male C57BL/6 mice (18-20 g) were purchased from the SIPPR/BK Lab Animal Ltd. (Shanghai, China), housed under SPF conditions with free access to food and water (temperature: 21 ± 2 °C and lighting: 8:00-20:00). Animal protocols were approved by Committee on Ethics of Biomedicine of Second Military Medical University (IACUC-2017324). All mice were treated humanely and with efforts to minimize suffering. To induce death with a minimum of pain and distress, all mice were euthanized by displacement of air with 100% carbon dioxide to collected tissue samples for further analyses. There were no surviving animals at the end of study.

Diabetes mellitus was induced in male C57BL/6 mice by streptozotocin (STZ; Amresco, Solon, Ohio) treatment. STZ was dissolved in 0.1 mM sodium citrate buffer (pH 4.5) and 60 mg/kg body weight was administered daily by intraperitoneal injection for 5 days (administration volume: 10 ml/kg; administration concentration: 6 mg/ml). Mice whole blood was obtained from the tail veins on day 20. Blood glucose levels were measured using a blood glucose monitoring system (Sinocare, Changsha, China). Mice with fasting blood glucose levels over 13.8 mM were defined as diabetic mice. Control mice were treated with citrate buffer (n = 8). STZ-induced diabetic mice (mentioned above) were randomly divided into two groups, each with 0.5% CMC-Na (n = 8) or metoprolol (Sigma-Aldrich, 100 mg/kg, intragastric administration (ig), n = 8) 14 days of treatment. The control mice received vehicle only. On day 34, mice were used for wound closure experiments or EPC isolation.

Western blot analysis

Samples of approximately 20 µg were run on 10% SDS-PAGE. The proteins were then electro-transferred to PVDF membranes (Millipore). The membranes were incubated in blocking buffer (Beyotime Biotechnology, China) for 4 h at room temperature. The blots were then incubated overnight at 4 °C with primary antibodies for eNOS, p-eNOS (Ser-1177) and β-Actin (Cell Signaling Technology, Danvers, Mass), and then incubated with HRP-conjugated secondary antibody (1:1000, Promega) (1:5,000; Cell Signaling Technology, Danvers, Mass) for 1 h at 25 °C. The Luminescence signal was obtained using GE Amersham AI600 (GE Healthcare), and the bands were quantified by Image J software (NIH, USA).

Measurement of wound closure and angiogenesis

Mice were anesthetized with isoflurane (3%), and the back was hairless and wiped 3 times with betaine and 75% ethanol before surgery. A six mm round wound was created with a biopsy punch. Each wound area was tracked every 2 days with a transparent, biocompatible transparent dressing (Johnson and Johnson, Arlington, Texas, USA) for 12 days to measure wound closure rate. The traces were digitized and the area was calculated using Image J software.

Lesions were obtained on days 3, 6, and 9 after wounding. Samples were fixed in paraformaldehyde before wax, then embedded in paraffin and sectioned at 5-µm intervals. Immerse the slide in a 3% hydrogen peroxide/methanol bath for 20 min and rinse with distilled water to block endogenous peroxidase. After treatment with normal rabbit serum for 30 min (Beyotime, Shanghai), the slides were incubated with anti-CD 31 antibodies (1:50; BD) for 60 min at room temperature and then incubated with Vectastain Elite ABC reagent (Vector Lab) 30 min, Nova Red (Vector Lab) 15 min. Slides were counterstained with hematoxylin (Beyotime, Shanghai) for 10 s, differentiated in a 1% glacial acetic acid aqueous solution, and rinsed in running tap water. Capillaries were recognized as tubular structures positive for CD31, and capillary density in the healing wounds was quantified.

Endothelial function testing in patients

Twelve healthy individuals (6 Male, 49 ± 7.5 years, BMI 22 ± 1.3 kg/m2) and 43 Type 2 diabetic patients (10 onset diabetic patients: 5 Male, 52.7 ± 7.6 years, BMI 26.3 ± 4.5; 19 treated with metoprolol: 9 Male, 66 ± 10.6 years, BMI 25 ± 2.9 kg/m2; 14 treated with bisoprolol: 2 Male, 53 ± 12.3 years, BMI 28 ± 4.7 kg/m2) were recruited. All had normal tests for hematologic, renal and liver function. None of the participants with type 2 diabetes had suffered from hypoglycemia in the preceding week before the study. None of the subjects practiced vigorous exercise. Treatments used in this study were metoprolol 100 mg twice daily or bisoprolol 5 mg once daily. All the participants were aware of this study and had provided written informed consent forms; this study was approved by the Committee on Ethics of Biomedicine of Second Military Medical University (SMMU-2017324).

The testing of endothelial function was performed as described previously (Axtell, Gomari & Cooke, 2010). Subjects were asked to quit smoking and refrain from drinking alcohol or caffeinated beverages for 12 h. During the test, the subject was seated in a particularly comfortable chair with both hands at the height of the heart. The Endo-PAT 2000 device (Itamar Medical Ltd, Caesarea, Israel) was used to obtain inter-finger pulsograms to record finger arterial pulse wave amplitude (PWA). Place a pneumatic probe on the index finger of each hand to record peripheral arterial tension (PAT). After a 20-minute equilibration period (temperature range 21−24 °C), baseline levels were measured for 5 min at rest and then for 5 min with one arm occluded. Obstruction is caused by inflating the pressure cuff of the upper arm to 50 mmHg above systolic blood pressure and then releasing it to induce reactive (blood flow mediated) congestion. Another un-occluded hand is used as a reference to correct potential systemic changes. The post-obstructive PWA was measured starting 90 s after cuff deflation, for 210 s. Endothelial function was calculated as the ratio of the average post-occlusion PWA to the average 5-minute baseline PWA and was corrected for systemic changes and baseline signal amplitude. The signal is analyzed using a computer-automatic algorithm to eliminate differences within and between observers.

Statistical analysis

Data are shown as the mean ± S.E.M. The statistical significance of differences between groups was obtained by the unpaired Student’s t-test or 1-way ANOVA with Newman-Keuls multiple comparison test in GraphPad Pro7.0 (GraphPad, San Diego, CA). Comparisons between multiple time points were analyzed by repeated-measurements analysis of variance with Bonferroni post-tests. Differences were considered to be significant at p < 0.05.

Results

Comparative effects of different β-blockers on tube formation and migration capacities in HUVECs and BM-EPCs

Five β-blockers (metoprolol, atenolol, propranolol, bisoprolol, and nebivolol) were included in this study. Propranolol and metoprolol increased tube formation capacity by 91 and 39% in HUVECs, respectively, compared with the control group (Figs. 1A–1F and 1M). Propranolol, metoprolol, and nebivolol also increased migration capacity by 66%, 93%, and 68% in HUVECs, respectively (Figs. 1G–1L and 1N). However, tube formation and migration capacities were not observed to increase significantly with the induction of atenolol or bisoprolol in HUVECs. Figures 1O–1BB shows that tube formation and migration capacities of BM-EPCs were increased significantly in the metoprolol group. Bisoprolol significantly increased tube formation capacity in BM-EPCs, while propranolol increased the migration capacity (Figs. 1AA and 1BB). Thus, metoprolol was shown to enhance tube formation and migration capacities in both HUVECs and BM-EPCs.

Figure 1 Effects of five β-blockers on HUVECs and EPCs tube formation and migration.

HUVECs or EPCs were incubated with PRO (propranolol, 0.3 μM), ATE (atenolol, 3 µM), MET (metoprolol, 0.3 µM), BIS (bisoprolol, 0.3 µM), NEB (nebivolol, 3 nM), or control for 24 hours and then tested. (A–L) Typical HUVECs tube formation and migration. (M–N) Quantitative evaluation of the tube numbers and ther numbers of migrated HUVECs from (A–L). (O–BB) Typical images and quantitative evaluation of the tube numbers and the numbers of migrated EPCs. Scale bar: 50 µm. n = 6. Data represent mean ± SEM. *, p < 0.05, **, p < 0.01, ***, p < 0.001 compared with unstimulated control.

Metoprolol improved BM-EPC function in diabetic mice

We also tested the effects of metoprolol on the capacities of tube formation and migration in BM-EPCs under a high glucose condition in vitro or in vivo. Tube formation and migration capacity were significantly reduced in the high glucose group compared with the control (Figs. 2A–2H). Treatment with metoprolol significantly improved BM-EPC function, compared with the high glucose treatment (Figs. 2A–2H).

Figure 2 Metoprolol (MET) improved EPC functions under high glucose condition (HG) or from streptozotocin-induced diabetic mice (STZ).

(A–D) Metoprolol enhanced the tube formation capacity of EPCs treated with high glucose (33 mM) for 24 h. (E–H) Metoprolol enhanced the migration capacity of EPCs treated with high glucose (33 mM) for 24 h. (I–L) Metoprolol enhanced the tube formation capacity of EPCs from diabetic mice. (M–P) Metoprolol enhanced the migration capacity of EPCs from diabetic mice. (Q–T) Metoprolol increased the number of circulating EPCs in diabetic mice. Scale bar: 50 µm. n = 6. Data represent mean ± SEM. **, p < 0.01, ***, p < 0.001 compared with unstimulated control. #, p < 0.05, ###, p < 0.001 compared with the HG group or STZ group.

Fifteen days after 5 days of treatment with STZ, the blood glucose levels of STZ-induced diabetic mice were significantly increased compared with the control group (Fig. S1). BM-EPC from STZ-induced diabetic mice showed significantly reduced cell migration capacity. Compared with untreated diabetic mice, metoprolol treatment significantly improved the tube formation ability and the migration ability of STZ-induced diabetic mice (Figs. 2I–2P). Similarly, the percentage of circulating EPC was significantly reduced in STZ-induced diabetic mice compared with that in the control group. Metoprolol treatment prevented this reduction in EPC in diabetic mice (Figs. 2Q–2T).

Metoprolol reduced superoxide generation and increased phospho-eNOS levels in BM-EPCs from diabetic mice

As shown in Figs. 3A–3D, superoxide levels in the high glucose-loaded BM-EPCs were significantly higher than those in the control. Metoprolol significantly reduced superoxide levels in BM-EPCs under the high glucose condition. Similarly, the enhanced superoxide anion production in BM-EPCs of STZ-induced diabetic mice was also inhibited by metoprolol (Figs. 3E–3H). The effects of five β-blockers on the high-glucose-induced generation of reactive oxygen species (ROS) were also compared in HUVECs (Fig. 4). Both the metoprolol and bisoprolol groups showed significantly lower O2−levels compared to the high glucose group. Treatment with propranolol, atenolol, or nebivolol alone had no obvious effect on O2− production under the high-glucose condition in HUVECs.

Figure 3 Metoprolol (MET) decreased intracellular O2- production and increased eNOS phosphorylation in EPCs.

(A–D) High glucose (HG, 33 mmol/L) treated EPCs were stain with DHE and assayed using flow cytometry. (E–H) Intracellular O2- production in EPCs from diabetic mice was measured by flow cytometry using DHE. (I–J) Western blot analyses were performed to determine changes of phosphorylated and total eNOS expression in EPCs from STZ-induced diabetic mice with or without metoprolol treatment. n = 4 − 6. Data represent mean ± SEM. **, p < 0.01 compared with unstimulated control. #, p < 0.05, ##, p < 0.01 compared with the HG group or STZ group.

Figure 4 Effects of five β-blockers on high-glucose induced reactive oxygen species (ROS) generation in HUVECs.

HUVECs were incubated with PRO (propranolol, 0.3 µM), ATE (atenolol, 3 µM), MET (metoprolol, 0.3 µM), BIS (bisoprolol, 0.3 µM), NEB (nebivolol, 3 nM), or control for 24 h and then stained with DHE (Dihydroethidium). HG (high glucose) group HUVECs were treated with 33 mM D-glucose for 24 h. Cells were read at 518/605 nm in a microplate reader and captured under the fluorescence microscope. Scale bar: 50 µm. n = 6. Data represent mean ± SEM. ***, p < 0.001 compared with unstimulated control. #, p < 0.05, ###, p < 0.001 compared with HG group.

Western blot analysis showed no differences in total eNOS protein expression in BM-EPCs among the groups (Figs. 3I and 3J). However, compared with the control group, the amount of phosphorylated eNOS relative to total eNOS was significantly reduced in BM-EPCs from STZ-induced diabetic mice. Metoprolol treatment significantly reversed the decrease in BM-EPCs from diabetic mice (Figs. 3I and 3J).

Metoprolol accelerated wound closure and angiogenesis in diabetic mice

Full thickness excisional skin wounds were generated on the backs of diabetic mice. These mice were treated with Metoprolol or vehicle, and examined every other day until day 12 (Fig. 5A). On day 12, it was noted that wound closure was significantly delayed in diabetic mice than in control mice (Figs. 5A–5V). However, the rate of wound closure was higher in the diabetic mice treated with metoprolol than in the untreated diabetic mice (Fig. 5A–5V).

Figure 5 Metoprolol (MET) accelerated wound closure rates in STZ-induced diabetic mice (STZ).

Wounds made with a 6 mm diameter biopsy punch were measured every 2 days until day 12. (A) Metoprolol improved the percentage of wound closure in STZ-induced diabetic mice compared with the untreated diabetic ones. (B–V) Representative photographs of the full thickness skin wounds. Mean ± SEM. n = 8. * p < 0.05 vs. Control; # p < 0.05 vs. STZ.

As angiogenesis plays a pivotal role in wound healing, we investigated whether the accelerated wound healing by metoprolol was associated with increased angiogenesis in wound tissues. At days 3, 6, and 9 after wounding, the number of microvessels was significantly smaller in wound beds of diabetic mice than in the controls (p < 0.05, Figs. 6A–6S). Compared with untreated diabetic mice, metoprolol treatment significantly increased capillary density in diabetic mice on days 6 and 9 (Figs. 6A–6S); metoprolol did not increase capillary formation on day 3.

Figure 6 Metoprolol (MET) stimulated angiogenesis in STZ-induced diabetic mice.

Biopsies were taken for immunohistology with CD31 mAb to detect blood vessels in the wound region (black arrows). Scale bar: 50 µm. Mean ± SEM. (* p < 0.05, ** p < 0.01; n = 5 per group).

Microvascular endothelial function

RH-PAT is a non-invasive technique that measures peripheral microvascular endothelial function by measuring changes in digital pulse volume during reactive hyperemia (Hamburg et al., 2008). The RH-PAT indices in onset diabetes patients were lower compared to those in the control subjects (1.6 ± 0.14 vs. 2.4 ± 0.18, p < 0.01; Figs. 7A, B). RH-PAT index was higher in metoprolol-treated patients compared with that in onset diabetic patients (2.3 ± 0.15 vs. 1.6 ± 0.14, p < 0.01; Figs. 7A, 7B). There was no significant difference in the RH-PAT indices of onset diabetic patients and bisoprolol-treated patients. The characteristics of the patient population can be found in Table 1.

Figure 7 Effect of metoprolol on endothelial function.

(A) Vascular endothelial function was evaluated by reactive hyperemia-peripheral arterial tonometry (RH-PAT) index in type 2 diabetes patients treated with bisoprolol or metoprolol. (B) Representative signals of RH-PAT in control or diabetic subjects. Normal response characterized by a distinct increase in the signal amplitude after cuff release compared with baseline. (* p < 0.05, ** p < 0.01).

Discussion

The present study demonstrated that metoprolol, a selective β1 receptor blocker, improved EPC function, accelerated angiogenesis, decreased the superoxide anion and increased the phosphorylation of eNOS in EPCs from diabetes. A validation study demonstrated that endothelial function was improved in diabetic patients treated with metoprolol. The results support the notion that the beneficial effects of metoprolol on endothelial and EPC function may be related to phosphorylation of eNOS and scavenging of superoxide anions.

These findings are important because most patients with diabetes and hypertension receive β-blocker treatment. Previous studies have shown that beta-blockers (such as propranolol) negatively regulate angiogenesis in ischemic models, such as hindlimb ischemia (7) and oxygen-induced retinopathy (18, 23). However, the issue remains controversial. Other studies have demonstrated that metoprolol and bisoprolol displayed proangiogenic activity in a mouse aortic ring model, which is independent of their ability to antagonize catecholamine action (Cheng et al., 2014; Stati et al., 2014). The beneficial effects of nebivolol beyond conventional β-blockers were also demonstrated in experimental models of post-myocardial infarction (Cheng et al., 2014; Stati et al., 2014). On the other hand, several reports were consistent with our findings, showing that metoprolol therapy improved endothelial function in patients with cardiac syndrome X (Majidinia et al., 2016) and increased the EPC proliferation in an acute myocardial infarction animal model (Stati et al., 2014). In this study, metoprolol significantly promoted angiogenesis both in vitro (cultured HUVECs and EPCs) and in vivo (wound healing in mice). Antihypertensive drugs and diabetic drugs are often combined in clinical practice. Yu et al. reported that metformin could also improve BM-EPC functions in STZ-induced diabetic mice. Dei et al. found that Vildagliptin, but not glibenclamide, increases circulating endothelial progenitor cell number in patients with type 2 diabetes. The combined impact of beta blockers and diabetic drugs in BM-EPCs function is also worth further study.

Table 1 Baseline Characteristics of the patients.

Characteristic	Control	Onset diabetic patients	Type 2 diabetic patients	
	subjects		Bisoprolol	Metoprolol	
n	12	10	14	19	
Age (years)	49 ± 7.5	53 ± 7.6	53 ± 12.3	66 ± 10.6	
Sex (male)	6	5	7	9	
BMI (Kg/m2)	22 ± 1.3	26 ± 4.5	28 ± 4.7*	25 ± 2.9*	
HbA1c (%)	5.9 ± 0.09	6.9 ± 0.46*	6.9 ± 0.50*	7.2 ± 0.38*	
TC (mmol/L)	4.3 ± 0.22	3.9 ± 0.36	4.1 ± 0.46	4.5 ± 0.27	
TG (mmol/L)	1.7 ± 0.24	1.6 ± 0.24	1.4 ± 0.23	1.9 ± 0.32	
HDL-C (mmol/L)	1.2 ± 0.09	1.4 ± 0.18	1.3 ± 0.08	1.1 ± 0.06	
LDL-C (mmol/L)	2.4 ± 0.15	2.4 ± 0.14	2.5 ± 0.18	2.4 ± 0.18	
Cardiovascular risk factors					
Arterial hypertension	0	10	14	19	
Hyperlipidemia	0	0	4	14	
Smoking	0	4	1	2	
Medical history					
Myocardial infarction	0	0	3	12	
Stroke	0	0	2	5	
Medications					
ACE inhibitor or AT2-blocker	0	0	7	12	
Calcium-channel blockers	0	0	6	7	
Diuretics	0	0	0	2	
Antiplatelet agents	0	0	3	10	
Metformin	0	0	1	5	
Sulfonylureas	0	0	3	5	
Insulin	0	0	0	2	
Notes.

Data are means ± SD or n .

* p < 0.05 vs. Control subjects.

Both type 1 and type 2 diabetic patients displayed fewer circulating EPCs and had impaired EPC function compared to the matched healthy subjects (De Vriese et al., 2000). Increased oxidative stress along with a subsequent decrease in eNOS phosphorylation contributes to EPC dysfunction in diabetes (Kolluru, Bir & Kevil, 2012). β-blockers have been mainly used based on their capacity to block the β-adrenoceptors (Gomes et al., 2006). However, part of the beneficial cardiovascular effects from β-blockers has been considered to be associated with the antioxidant properties (Haas et al., 2003). Gomes et al. (2006) showed that β-blockers (atenolol, labetalol, metoprolol, and propranolol et al.) are good ROS and/or RNS scavengers, which may be useful in preventing the oxidative damages. In the present study, the concentration of superoxide anion in the diabetic model was markedly reduced by metoprolol. Metoprolol significantly increased HG-induced eNOS dephosphorylation in EPCs. These results suggest that the effects of metoprolol on improving EPC function might be associated with the reduction of ROS generation and an increase in eNOS phosphorylation in diabetes or induced by HG.

The RH-PAT index calculated using the PAT signal is applied to a parameter of endothelial function. A low RH-PAT index is used to diagnose a patient with endothelial dysfunction (Bonetti et al., 2004). Thus, PAT is considered to be a useful, noninvasive examination for the prediction of cardiovascular events (Rubinshtein et al., 2010). Endothelial dysfunction, as measured by RH-PAT, was also present in diabetic patients (Pareyn et al., 2013). In this study, we compared the effects of chronic therapy with metoprolol and bisoprolol in diabetic patients. The average RH-PAT index was significantly higher in diabetic patients treated with metoprolol compared with that in patients treated with bisoprolol. This effect on endothelial function is predicted to be an intrinsic property of metoprolol. Therefore, conceivably, the beneficial effects of metoprolol in patients with hypertension and diabetes may be due to its preservation of normal endothelial function. However, age-related and gender-related differences in endothelial dysfunction should be considered in this study. Compared with men, endothelial dysfunction occurs late in women (Juonala et al., 2008). In our study, younger age and all-male gender may also partly account for the higher RH-PAT indexes in the control group.

This study had several limitations. The first is due to the relatively small sample size. There is a possibility that a significant difference in the RH-PAT index among different β-blockers may be demonstrated with a larger number of subjects. Second, except for metoprolol, the effects of other β-blockers on endothelial function under diabetic conditions need further detailed experiments. Third, the structure and function of the anti-oxidation effect from treatment with β-blockers also need further detailed experimentation.

Conclusions

Our work demonstrated that metoprolol could improve EPC function that is damaged by HG or in STZ-induced diabetic mice, accelerate wound healing in diabetic mice, and maintain microvascular endothelial function in diabetic patients. These results suggest a beneficial effect of metoprolol in the treatment of patients with diabetes mellitus combined with hypertension.

Supplemental Information

Supplemental Information 1 Changes of blood glucose concentration and body weight of streptozotocin-induced diabetic mice (STZ)

Blood glucose (A) and body weight (B) changes of STZ (60 mg/kg 5d, i.p.) treated mice, which was defined as diabetic mice (blood glucose value > 13.8 mmol/L) 15 days after STZ treatment. Values are mean SEM. n = 6 per group. * p < 0.05 vs. Control.

Click here for additional data file.

Supplemental Information 2 Western blot raw data

Click here for additional data file.

Additional Information and Declarations

Competing Interests

Author Contributions

Human Ethics

Animal Ethics

Data Availability

The authors declare there are no competing interests.

Lang Yan and Jiang-bo Zhu conceived and designed the experiments, prepared figures and/or tables, authored or reviewed drafts of the paper, and approved the final draft.

Yi-fan Dong and Ji-kuai Chen conceived and designed the experiments, prepared figures and/or tables, and approved the final draft.

Tao-lin Qing, Wen-jing Shi and Jin-feng Li analyzed the data, prepared figures and/or tables, and approved the final draft.

Ya-ping Deng and Xue Han performed the experiments, prepared figures and/or tables, and approved the final draft.

Fang-yuan Gao analyzed the data, authored or reviewed drafts of the paper, and approved the final draft.

Xiao-fang Zhang conceived and designed the experiments, authored or reviewed drafts of the paper, and approved the final draft.

Yi-jun Tian and Xiao-yu Dai performed the experiments, authored or reviewed drafts of the paper, and approved the final draft.

The following information was supplied relating to ethical approvals (i.e., approving body and any reference numbers):

All the participants were aware of this study and had provided written informed consent forms; this study was approved by the Committee on Ethics of Biomedicine of Second Military Medical University (SMMU-2017324).

The following information was supplied relating to ethical approvals (i.e., approving body and any reference numbers):

Animal protocols were approved by the Committee on Ethics of Biomedicine of Second Military Medical University (IACUC-2017324).

The following information was supplied regarding data availability:

The raw measurements are available in a Supplemental File.

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
