# Peer review of "Metoprolol rescues endothelial progenitor cell dysfunction in diabetes"

_PeerJ, doi:10.7717/peerj.9306_

## Round 0.1 · original submission · Minor Revisions

When revising your manuscript, please consider all issues mentioned in the comments from the three reviewers carefully and provide suitable responses for any comments. Please note that your revised submission may need to be re-reviewed.

PeeJ values your contribution and I look forward to receiving your revised manuscript.

Reviewer 1 ·

Basic reporting

no comment

Experimental design

no comment

Validity of the findings

no comment

Additional comments

In this paper, experiments were performed in cells, mice and clinical patients. The authors, at last, made a conclusion that metoprolol could rescue endothelial progenitor cell dysfunction in diabetes. The work of this paper is practical and the results of study have clinical significance. However, there are some problems to be further improved as well.
1. There is at least one spelling error in the manuscript, such as, in line 394, Figure 1, “ther” would be “their”. Please check the manuscript carefully.
2. The abscissas of Fig.2A and Fig.2C are lost. Based on customary reading approach reading from left to right and top to bottom , the authors ought to post the abscissas of Fig.2B and Fig.2D on the top of photographs. The graphic standards of Fig.2 should keep consistent with that of Fig.1.
3. In Fig.1, the effects of five beta-blockers on HUVECs tube formation and migration are showed by photographs and bar charts (Fig.1A and Fig.1B). However, the effects of five beta-blockers on BM-EPCs tube formation and migration are only displayed in bar charts without photographs. Please replenish relevant photographs.
4. The concentration unit should remain the same in paper, such as “mmol/L” and “mM”.
5. In Fig.2, the border sizes of bar charts between HG-MET- group and HG+MET- group are different . This mistake also can be found in Fig.3B. Besides, in Fig.4, the bar of CON group has border while that of another groups are without borders. Please check it carefully.
6. In Fig.6, authors used black arrows to note. However, authors didn’t explain what that mean in figure legends. The arrows may merit a brief explanatory note.
7. Optimum concentration of five beta-blockers need further detailed experiments.

Annotated reviews are not available for download in order to protect the identity of reviewers who chose to remain anonymous.

Reviewer 2 ·

Basic reporting

The MS entitle “Metoprolol rescues endothelial progenitor cell dysfunction in diabetes” addressed the endothelial dysfunction in diabetic patents, and how treatment of β-blockers, especially metoprolol, overcome the endothelial dysfunction in diabetic mice and patients. To address this, authors were used both in vitro and in vivo experimental methods and their scientific contributions are encouraging and useful further in-depth mechanistic investigation. The scientific contents, methodology, figures, tables and language are of good standards. However, some typo errors and queries should be addressed before its acceptance.

1. Is the reported work is carried out in combination with diabetic drugs (e.g. Metformin etc.)?

2. If not what could be the effect of these drugs on EPC function.

3. What is the concentration and volume of STZ was used to induce diabetic in mice? Should be mentioned in the methodology and results sections.

4. The drug names should be in lowercase letters throughout the MS except at starting of sentence.

5. In page 13, line no 171: The sentence “Measurement of PWA 210 s……” should be revised. It’s not meaningful.

6. In page 25, Figure 1 legend: “Quantitative evaluation of the tube numbers and ther numbers of migrated HUVECs from (A).” In this sentence the typo error ther should replaces with the.

Experimental design

The experimental design and the research work carried are within the scope & aim this journal. The hypothesis is well defined, and relevant work is carried with standard experimental methods. Although, the sample size (for in vivo work) is less, the outcome results are helpful for in-depth mechanistic investigations.

Validity of the findings

The obtained results are valid, statistically robust and clearly stated in the results and discussion section.

Additional comments

The scientific work and contribution is encouraging. However, author should have carried out their work using β-blockers in combination with diabetic drugs (e.g. metformin etc.) for better understanding the EPC functions.

Reviewer 3 ·

Basic reporting

no comment

Experimental design

no comment

Validity of the findings

no comment

Additional comments

This present manuscript demonstrates metoprolol can accelerate wound healing in diabetic mice and improve endothelial function in diabetic subjects possibly by improving the impaired EPC function.

The results of this research will be helpful for clinical therapeutics of diabetes vascular lesion. The manuscript can be accepted after some data supplemented.

1. In figure 1, the authors need to provide both photos and statistic data for EPCs, like they do on HUVEC.
2. Tube formation and migration are the main evaluation methods used in this research for experiments in vitro. In figure 2, the authors only provide the migration data for EPCs from STZ, they should supplement data of tube formation for these cells.
3. In figure 7, the clinic data illustrates the effect of metoprolol on endothelial function. However, diabetes treated with placebo should be an important control group and the data need to be provided.

---

## Round 0.2 · accepted · Accept

The authors have responded well to the comments of the three reviewers.

Reviewer 1 ·

Basic reporting

no comment

Experimental design

no comment

Validity of the findings

The work of this paper is practical and the results of study have clinical significance.

Additional comments

The author has made a serious revision according to the opinions of the reviewers, so it is suggested to accept this paper.

Reviewer 2 ·

Basic reporting

No comment

Experimental design

No comment

Validity of the findings

No comment

Additional comments

No comments